# Critical non-Hermitian skin effect

Linhu Li [1✉], Ching Hua Lee [1✉], Sen Mu[1✉] & Jiangbin Gong [1✉]

Critical systems represent physical boundaries between different phases of matter and have been intensely studied for their universality and rich physics. Yet, with the rise of non-Hermitian studies, fundamental concepts underpinning critical systems - like band gaps and locality - are increasingly called into question. This work uncovers a new class of criticality where eigenenergies and eigenstates of non-Hermitian lattice systems jump discontinuously across a critical point in the thermodynamic limit, unlike established critical scenarios with spectrum remaining continuous across a transition. Such critical behavior, dubbed the "critical non-Hermitian skin effect", arises whenever subsystems with dissimilar non-reciprocal accumulations are coupled, however weakly. This indicates, as elaborated with the generalized Brillouin zone approach, that the thermodynamic and zero-coupling limits are not exchangeable, and that even a large system can be qualitatively different from its thermodynamic limit. Examples with anomalous scaling behavior are presented as manifestations of the critical non-Hermitian skin effect in finite-size systems. More spectacularly, topological in-gap modes can even be induced by changing the system size. We provide an explicit proposal for detecting the critical non-Hermitian skin effect in an RLC circuit setup, which also directly carries over to established setups in non-Hermitian optics and mechanics.

[1] Department of Physics, National University of Singapore, Singapore 117542, Singapore. ✉email: rubilacxelee@gmail.com; phylch@nus.edu.sg; senmu@u.nus.edu; phygj@nus.edu.sg

Lying at the boundary between distinct phases, critical systems exhibit a wide range of interesting universal properties from divergent susceptibilities to anomalous scaling behavior. They have broad ramifications in conformal and statistical field theory[1–4], Schramm–Loewner evolution[5,6], entanglement entropy (EE)[7–14], and many other contexts. Recently, concepts crucial to criticalities—like band gaps and localization—have been challenged by studies of non-Hermitian systems[15,16] exhibiting exceptional points[17–27] or the non-Hermitian skin effect (NHSE), which are characterized by enigmatic bulk-boundary correspondence (BBC) violations, robust-directed amplifications, discontinuous Berry curvature, and anomalous transport behavior[28–40].

We uncover here a class of criticality, dubbed the "critical non-Hermitian skin effect (CNHSE)", where the eigenenergies and eigenstates in the thermodynamic limit "jump" between different skin solutions discontinuously across the critical point. This is distinct from previously known phase transitions (Hermitian and non-Hermitian) (Fig. 1), where the eigenenergy spectrum can be continuously interpolated across the two bordering phases. A CNHSE transition, by contrast, is characterized by a discontinuous jump between two different complex spectra along with two different sets of eigenstates. As elaborated below, this behavior appears generically whenever systems of dissimilar NHSE localization lengths are coupled, no matter how weakly. Importantly, at experimentally accessible finite system sizes, the jump smooths out into an interpolation between the two phases in a strongly size-dependent manner, such that the system may exhibit qualitatively different properties, i.e., real vs. complex

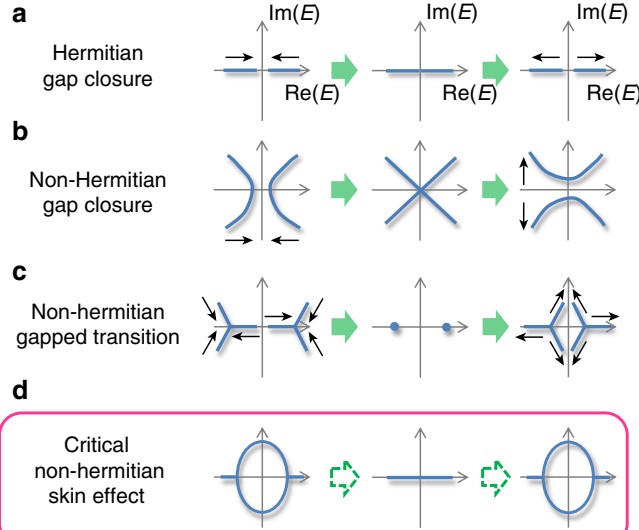

**Fig. 1 Four different types of critical transitions.** Hermitian phase transitions (**a**) are marked by gap closures along the real line. In non-Hermitian cases (2nd to 4th rows, axis labels omitted), spectral phase transitions can take more sophisticated possibilities in the 2D complex energy plane. For instance, the spectral topology can change under line gap closures (**b**) or shrink continuously to a point and re-emerge in a different topological configuration (**c**), without the gap ever closing[38]. The spectrum continuously passes through a gapless or point-like regime in the first three cases, as indicated by the black arrows. The critical non-Hermitian skin effect (**d**), however, is special in that OBC spectrum in the thermodynamic limit, denoted $E_\infty$, jumps discontinuously from one configuration (left), to a different configuration (middle), and to another (right) as certain parameter changes from $-\epsilon$ to 0 (critical border), and to $\epsilon$, for an arbitrarily small $\epsilon$, without ever interpolating between the configurations even though the parameter is continuously tuned.

spectrum or presence/absence of topological modes at different system sizes. Being strongly affected by minute perturbations around the critical point, such behavior may prove useful in sensing applications[41,42].

## Results

**Hints of the critical non-Hermitian skin effect from the general Brillouin zone.** In non-Hermitian systems with unbalanced gain and loss, the spectra under periodic boundary conditions (PBCs) and open-boundary conditions (OBCs) can be very different[28,29,31,43–45]. Indeed, under OBC, eigenstates due to NHSE can exponentially localize at a boundary, in contrast to Bloch states under PBCs. This also explains the possible violation of the BBC, taken for granted in Hermitian settings.

The celebrated GBZ formalism aims to restore the BBC through a complex momentum deformation[29–31,36–38]. Rigorously applicable for bounded but infinitely large systems, it has however been an open question whether the GBZ can still accurately describe finite-size systems. The GBZ of a momentum-space Hamiltonian $H(z)$, $z = e^{ik}$ can be derived from its characteristic Laurent polynomial (energy eigenequation)

$$f(z, E) := \det[H(z) - E] = 0, \quad (1)$$

where $E$ is the eigenenergy. While the ordinary BZ is given by the span of allowed real quasimomenta $k$, the GBZ is defined by the complex analytically continued momentum $k \rightarrow k + i\kappa(k)$, with the NHSE inverse decay length $\kappa(k) = -\log|z|$ determined by the smallest complex deformation $z \rightarrow e^{ik}e^{-\kappa(k)}$ such that $f(z, E)$ possesses a pair of zeros $z_\mu, z_\nu$ satisfying $|z_\mu| = |z_\nu|$ for the same $E$[29,31,38]. Due to the double degeneracy of states with equal asymptotic decay rate at these $E$, there exist a pair of eigenstates $\psi_\mu, \psi_\nu$ that can superpose to satisfy OBCs, i.e., zero net amplitude at both boundaries. As such, provided that the characteristic polynomial is not reducible, the OBC spectrum in the thermodynamic limit (denoted as $E_\infty$) can be obtained from the PBC spectrum via $E(e^{ik}) \rightarrow E(e^{ik}e^{-\kappa(k)})$, apart from isolated topological modes. Thus it is often claimed that the BBC is "restored" in the GBZ defined by $k \rightarrow k + i\kappa(k)$ or, at the operator level, with the surrogate Hamiltonian $H(e^{ik}) \rightarrow H(e^{ik}e^{-\kappa(k)})$[38]. In general, different $E$ (energy band) solutions can admit different functional forms of $\kappa(k)$, leading to band-dependent GBZs that have recently also been described with the auxiliary GBZ formalism[37]. Since $e^{ik}e^{-\kappa(k)}$ is generically non-analytic, it represents effectively non-local hopping terms[38]. As such, the GBZ description challenges the very notion of locality, which is central to critical systems, by effectively "unraveling" the real-space eigenstate accumulation through replacing local hoppings with effectively non-local ones.

Due to the robustness of the NHSE, eigenspectra predicted from the GBZ typically are approached rapidly by the exact numerically obtained OBC spectra even for small system sizes ($\mathcal{O}(10^1)$ sites). In principle, the convergence should be exact in the thermodynamic limit, but in practical computations, floating-point errors $\epsilon_0$ are continuously amplified as they propagate across the system. We hence expect accurate numerical spectra only when $L < -\log(\epsilon_0)/\max(\kappa)$, a condition always checked to be satisfied here to ensure that physical phenomena presented below are not due to numerical errors common in computations with non-reciprocal systems. However, the numerical agreement in eigenspectra between finite-size systems and the GBZ predictions fails spectacularly near a critical point where $f(z, E)$ changes from being reducible to irreducible. To understand the significance of this algebraic property of reducibility, consider a set of coupled irreducible subsystems

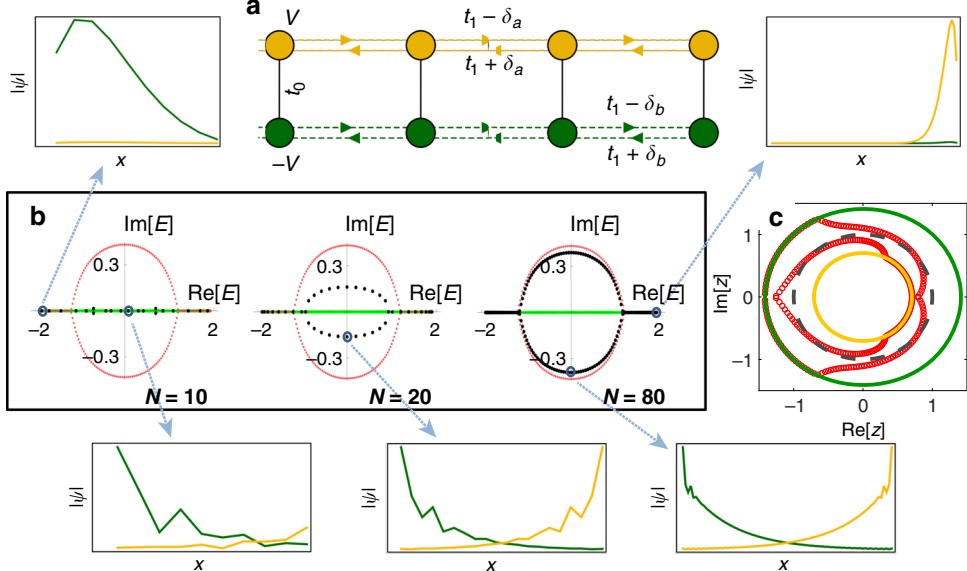

**Fig. 2 Critical non-Hermitian skin effect and finite-size scaling in the model of two coupled Hatano–Nelson chains. a** The two-chain model [Eq. (3)] with hopping asymmetry in chains $a$, $b$ denoted by $\delta_{a/b}$, and on-site energy offset $\pm V$. A small inter-chain $t_0$ can cause significant coupling when $\delta_a \neq \delta_b$. **b** Open-boundary spectra (black dots) and eigenstate profiles (insets) with $N = 10$, $20$, and $80$ unit cells and the coupling parameter $t_0 = 0.01$, showing very different spectral behavior at different system sizes $N$. At small $N \approx 10$, coupling effects are negligible, with the spectrum coinciding with the real-value open-boundary $E_\infty$ spectrum (green) in the decoupled thermodynamic limit. As $N$ increases, the spectrum gradually approaches the open-boundary $E_\infty$ spectrum (red) for the coupled thermodynamic limit, with hybridization becoming sharper. **c** The generalized Brillouin zones of the two chains in the decoupled limit (yellow and green), and that of the two bands with $t_0 = 0.01$ (red circles, numerically obtained at $N = 80$). The Brillouin zone with $|z| = 1$ is given by the gray dash circle. Other parameters are $t_1 = 0.75$, $\delta_a = -\delta_b = 0.25$, and $V = 0.5$.

described by the characteristic polynomial

$$f(z, E) = f_0 + \prod_i f_i(z, E), \qquad (2)$$

where $f_i(z, E)$ is the characteristic polynomial of the $i$-th subsystem, and $f_0$ is a constant that represents the simplest possible form for the subsystem coupling. When $f_0 = 0$, $f(z, E)$ completely factorizes into irreducible polynomials, as expected from a Hamiltonian $H(z)$ that block-diagonalizes into irreducible sectors associated with the individual $f_i(z, E)$'s. In particular, the OBC spectrum of this completely decoupled scenario is derived from the independent $\kappa_i(k)$'s of each subsystem, each determined by $z_\mu$, $z_\nu$ from the same subsystem.

Yet, a nonzero coupling $f_0$, no matter how small, can have marked physical consequences by hybridizing different sectors of $f_i$ significantly. Indeed, such hybridization is inevitable in the thermodynamic limit, with OBC eigenstates formed from superpositions of eigenstates $\psi_\mu$, $\psi_\nu$ from dissimilar subsystems, each corresponding non-Bloch momenta $-i \log z_{\mu/\nu}$. Hence the GBZs, i.e., $\kappa(k)$'s of the coupled system, which are defined in the thermodynamic limit, are thus determined by all pairs of $|z_\mu| = |z_\nu|$ not necessarily from the same subsystem. Therefore, the GBZs in the coupled case, no matter how small is $f_0$, can differ from the decoupled GBZs at $f_0 = 0$. That is, the thermodynamic limit and the $f_0 \to 0$ limit are not exchangeable. However, since an actual finite physical system cannot possibly possess very different spectrum and band structure upon an arbitrarily small variation in its system parameter, the GBZ picture becomes inapplicable when describing finite systems (small or large) in the presence of CNHSE.

**Anomalous finite-size scaling from CNHSE.** For illustration, we turn to a minimal example of two coupled non-Hermitian 1D

Hatano–Nelson chains[46] each containing only non-reciprocal (unbalanced) nearest neighbor (NN) hoppings (Fig. 2a). Its Hamiltonian reads as

$$H_{2-chain}(z) = \begin{pmatrix} g_a(z) & t_0 \\ t_0 & g_b(z) \end{pmatrix}, \qquad (3)$$

with $g_a(z) = t_a^+ z + t_a^-/z + V$ and $g_b(z) = t_b^+ z + t_b^-/z - V$, $t_{a/b}^\pm = t_1 \pm \delta_{a/b}$ being the forward/backward hopping of chains $a$ and $b$. This model can be also realized with a reciprocal system with the NHSE in a certain parameter regime (Supplementary Note 1). When $t_0 = 0$, the two chains are decoupled, and the characteristic polynomial is reducible as $f(z, E) = [g_a(z) - E][g_b(z) - E]$. Each factor $f_{a/b}(z, E) = g_{a/b}(z) - E$ determines the skin eigensolutions of its respective chain. However, even an infinitesimal coupling $t_0 \neq 0$ generically makes $f(z, E)$ irreducible, providing that the two chains correspond to different GBZs (see "Methods" section). Specifically, consider the simple case of $t_a^+ = t_b^- = 1$ and $t_a^- = t_b^+ = 0$. Without couplings ($t_0 = 0$), the two chains under OBC respectively yields a Jordan-block Hamiltonian matrix in real space, with the spectrum given by $E = \pm V$. Because the eigenstates of the decoupled chains are exclusively localized at the first or the last site, their GBZs collapse[45]. By contrast, for any $t_0 \neq 0$, $f(z, E) = E^2 - E(z + z^{-1}) + (z + V)(z^{-1} - V) - t_0^2$ is irreducible (here $-t_0^2 = f_0$ from Eq. (2)), insofar as the eigen-energy roots $E = \cos k \pm \sqrt{t_0^2 + (V + i \sin k)^2}$ are no longer Laurent polynomials in $z = e^{ik}$ that can be separately interpreted as de facto subsystems with local hoppings. In fact, in higher degree polynomials, an algebraic expression for $z$ may not even exist as implied by the Abel–Ruffini theorem. Importantly, the corresponding OBC $E_\infty$ spectrum and the GBZ for $t_0 \neq 0$ are now qualitatively different. As derived in the Methods section, setting $|z_a| = |z_b|$ gives OBC spectrum (in the thermodynamic limit):

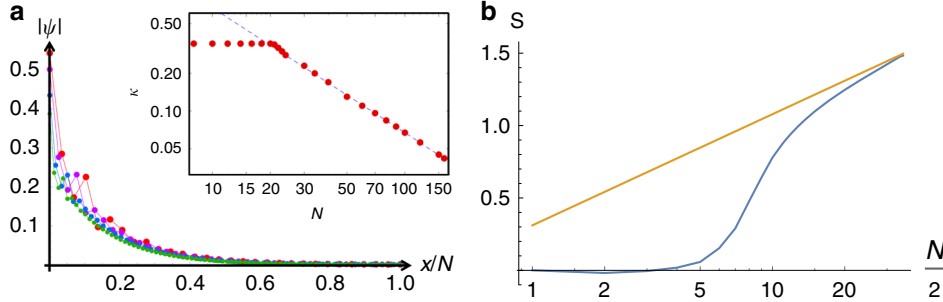

**Fig. 3 Anomalous scaling behaviors of the critical non-Hermitian skin effect. a** Scale-free open-boundary skin eigenstate of the largest Im[$E$] eigenenergy of $H_{2\text{-chain}}$ at system sizes $N = 20, 40, 60$, and 80 (red, purple, blue, green, respectively). Its rescaled profile, despite decaying exponentially rather than power-law, remains invariant across different $N$. This scale invariance persists in the $N > 20$ regime, and is due to the $N^{-1}$ decay (dashed line) of the inverse skin depth (red dots), as plotted in the inset. Parameters follow Fig. 2's, except with $t_0 = 10^{-3}$. **b** Entanglement entropy $S$ (blue) of a half-filled open-boundary $H_{2\text{-chain}}$ at odd system sizes $N$, with real-space cut at $\lfloor \frac{N}{2} \rfloor$ (the floor function of $N/2$) and parameters $t_1 = 0.58$, $V = 1$, $t_0 = 0.4$, and $\delta_a = -\delta_b = 0.25$. It saturates near zero in the gapless decoupled small $N$ regime, but scales like $\sim \frac{1}{3} \log N$ (yellow) in the gapless coupled large $N$ regime.

$E^2_\infty = \frac{1-\eta^2}{1+\eta^2} + V^2 + t_0^2 \pm 2\sqrt{t_0^2 - \eta^2 + \eta^2 t_0^2}/(1+\eta^2)$, with $\eta \in \mathbb{R}$. Clearly, even one now takes the $t_0 \to 0$ limit, $E^2_\infty$ only simplifies to $E^2_\infty \to V^2 + \frac{1 \pm i\eta}{1 \mp i\eta}$, which is not the above-mentioned OBC spectrum of the two decoupled chains. Likewise, the $t_0 \to 0$ limit of the coupled GBZ, which can be shown to be the locus of $z = \pm\sqrt{V^2 + e^{i\theta}} - V$ and $z = 1/[V \pm \sqrt{V^2 + e^{i\theta}}]$, $\theta \in [0, 2\pi]$, has nothing in common with the collapsed GBZs of the decoupled case.

This paradoxical singular behavior of GBZs leads to anomalous scaling behavior in finite-size systems that are more relevant to experimental setups. The discontinuous critical transition illustrated above becomes a smooth crossover between the different OBC $E_\infty$ solutions. As the size $N$ of a coupled system is varied, its physical OBC spectrum interpolates between the decoupled and coupled OBC $E_\infty$ solutions. As illustrated in Fig. 2b for the two-chain model Eq. (3) at small coupling $t_0 = 0.01$ (with $t_1 = 0.75$ and $\delta_a = -\delta_b = 0.25$ for well-defined skin modes), the OBC spectrum (black dots) changes markedly from $N = 10$ to 80 unit cells. For small $N = 10$, the spectrum approximates the OBC $E_\infty$ (green) for $t_0 = 0$ (which lies on the real line), with the associated GBZs given by two perfect circles in the complex plane (Fig. 2c). At large $N = 80$, the spectrum converges toward the true OBC $E_\infty$ (red curve) with nonzero coupling, where the associated respective GBZs of the two bands (also shown in Fig. 2c) are much different from the two circles as decoupled GBZs. Indeed, the eigenstates for $N = 10$ are almost entirely decoupled across the two chains, while those for $N = 80$ are maximally coupled/decoupled depending on whether they approach the red/green $E_\infty$ curves. In the intermediate $N = 20$ case, the OBC spectrum lies far between the two $E_\infty$'s, and cannot be characterized by their GBZs. The size-dependent behavior of the OBC spectrum is further elaborated through a spectral-flow study[31] in Supplementary Note 2.

Let us now explain the above-observed marked size-dependent spectra via the competition between dissimilarly accumulated skin modes and the couplings across them. The general conditions for such are unveiled in the "Methods" section. In our model (Eq. 3), the inverse decay lengths in chains $a$, $b$ are given by $\kappa_{a/b} = \frac{1}{2} \log(t^+_{a/b}/t^-_{a/b})$, which will be dissimilar as long as $\delta_a \neq \delta_b$. After performing a similarity transformation that rescales each site $j$ by a factor of $e^{j\kappa_b}$, chain $b$ becomes reciprocal with $\kappa'_b = 0$ while chain $a$ has a rescaled inverse decay length $\kappa'_a = \kappa_a - \kappa_b$. If $\kappa'_a \neq 0$, chain $a$ always possesses exponentially growing skin modes scaling like $e^{\kappa'_a N}$ at one end. As such, the

coupling $t_0$, even if being extremely small, still affects the spectrum and eigenstates markedly as the system size $N$ increases, as further elaborated in the "Methods" section.

**Scale-free exponential wavefunctions.** A hallmark of conventional critical systems is scale-free power-law behavior, particularly in the wavefunctions. Interestingly, such scale-free behavior can also be found in the exponentially decaying wavefunctions, i.e., skin modes. Shown in Fig. 3a are the profiles of the slowest decaying eigenstates $\psi(x)$ of $H_{2\text{-chain}}$ at different system sizes $N = 20, 40, 60$, and 80, with the horizontal axis normalized by $N$. These featured eigenstates belong to the top of the central black ring in Fig. 2b, with their distance from the coupled OBC $E_\infty$ ring (red) decreasing as $\sim N^{-1}$. Unlike usual exponentially decaying wavefunctions with fixed spatial decay length, here $|\psi(x)| \sim e^{-\kappa x}$ with $\kappa \sim N^{-1}$ (Fig. 3b), such that the overall profile $\psi(x)$ has no fixed length scale. Such unique scale-free eigenmodes result from the slow critical migration of the eigenstates between $E_\infty$ solutions (Fig. 2a, inset).

**Anomalous correlations and entanglement entropy.** The OBC spectra can be gapped for certain system sizes where EE obeys an area law scaling, and then become gapless at other sizes where the EE scaling is replaced by a logarithmic dependence on system size[47]. This indicates that the CNHSE can lead to an unusual scaling behavior of the EE. Consider for instance the OBC $H_{2\text{-chain}}$ (Eq. 3) with parameters chosen to gap out the OBC spectrum at small system sizes $N$. With all Re[$E$] < 0 states occupied by spinless-free Fermions, the real-space entanglement entropy $S$ (blue curve in Fig. 3b) exhibits a crossover from the decoupled gapped regime at $N \leq 5$ where it remains a constant due to the size-independent area of boundaries (two ends), to the gapless regime $N > 20$ where it approaches the $\frac{1}{3} \log N$ behavior of a gapless system (yellow line). In generic CNHSE scenarios with multiple competing OBC $E_\infty$ loci, $S$ can scale differently at different system size regimes, choices of fillings, and entanglement cuts, challenging the notion of single well-defined scaling behavior. As further shown in Supplementary Note 3, the two-Fermion correlator $\langle \psi(1)\psi(x) \rangle$ characterizing the EE also crosses over from rapid exponential decay at small $N$ to $1/x$ power-law decay at large $N$. Remarkably, the probability of finding another Fermion nearby generally increases drastically when the system is enlarged (with filling fraction maintained).

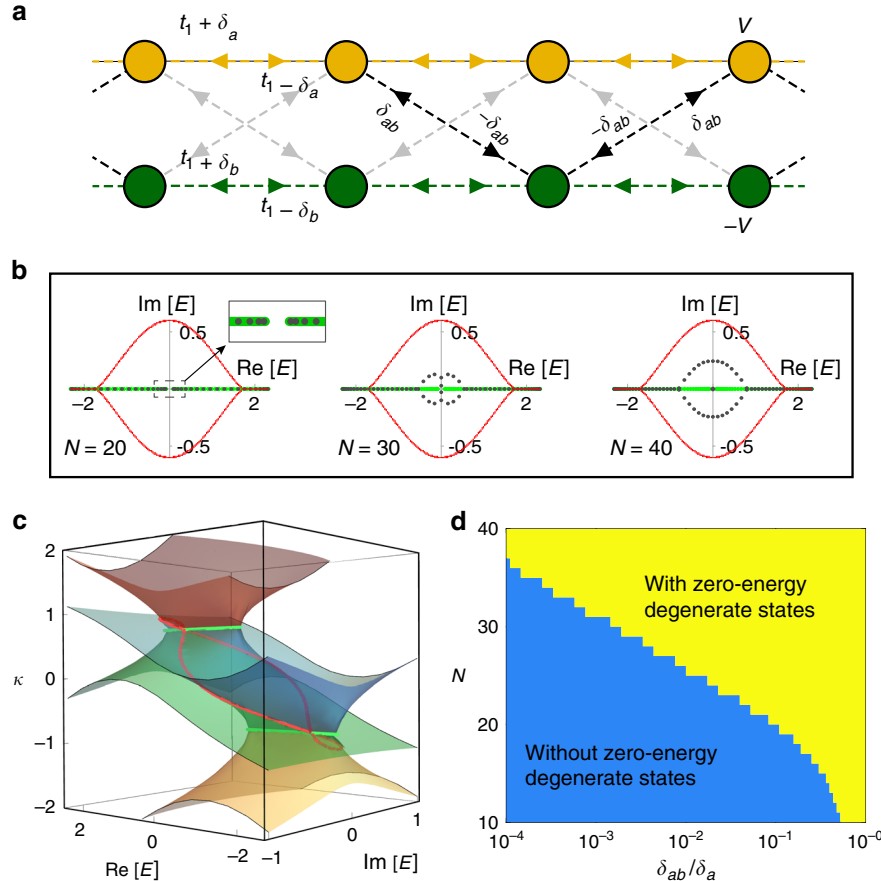

**Fig. 4 Topological transition induced by changing the system's size. a** Sketch of the $H_{CNHSE-SSH}$ model with cross inter-chain non-reciprocal couplings $\pm\delta_{ab}$. **b** Open-boundary spectra (black dots) at $N = 20$, 30, and 40 unit cells and coupling $\delta_{ab} = 0.5 \times 10^{-3}$. The main part of the spectrum in each case behaves similarly as the model in Fig. 2b, except for a pair of topological edge states blue emerging within the point gap at zero energy. The open-boundary $E_\infty$ spectrum is given by green and red colors in the decoupled and coupled thermodynamic limits respectively. Other parameters are $\delta_a = -\delta_b = 0.5$, $t_1 = 0.75$, and $V = 1.2$. **c** $\kappa$ solutions (red, blue, green, and yellow surfaces) of $f(z, E) = 0$ as a function of the complex energy, with the same parameters in **b**. Intersecting regions (green and red dotted lines) give the open-boundary skin solutions of the system in the thermodynamic limit. Among them, green lines correspond to the skin solutions of two decoupled chains at $\delta_{ab} = 0$. The solutions of red curves emerge at a small but nonzero $\delta_{ab}$, and the skin solutions of the weakly coupled system are given by the intersecting regions with the smallest $|\kappa|$, i.e., the red loop in the center and green lines at the two ends with large and small Re[$E$]. **d** Emergence of in-gap degenerate modes as a function of $\delta_{ab}/\delta_a$ and $N$ with $\delta_a = -\delta_b = 0.5$, $t_1 = 0.75$, and $V = 1.2$, with the plotted boundary scaling logarithmically with $N$.

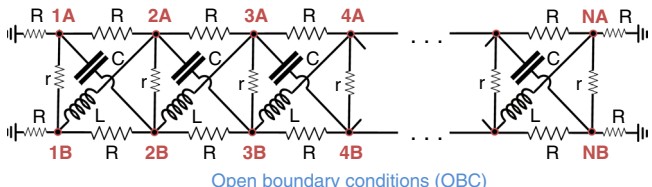

Open boundary conditions (OBC)

**Fig. 5 A circuit lattice realizing the critical non-Hermitian skin effect.** The Laplacian of it is given by Eq. (5) with $N$ unit cells with open-boundary conditions. The nodes $A$ and $B$ of unit cells 1, 2, 3, ... are labeled accordingly. Grounded $R$ resistors are added to make the boundary nodes experience the same outdegree.

**Size-dependent topological modes**. Topological modes are usually associated with bulk invariants in the thermodynamic limit, with finite-size effects having a diminishing role in the face of topological robustness. This intuition is not necessarily true in non-Hermitian systems, as hinted from ref. [34], where an infinitesimal instability can cause a $Z_2$ topological transition in the thermodynamic limit[34]. Remarkably, the CNHSE here can cause topological edge modes to appear only at certain system size

regimes. Consider replacing the non-reciprocal intra-chain couplings of our $H_{2\text{-chain}}$ model with inter-chain couplings with non-reciprocity $\pm\delta_{ab}$ between adjacent unit cells (Fig. 4a), as described by the following CNHSE Su–Schrieffer–Heeger (SSH) model[48]:

$$H_{CNHSE-SSH}(z) = h_y(z)\sigma_y + h_z(z)\sigma_z + h_0(z)\mathbb{I}, \quad (4)$$

where $h_y(z) = i\delta_{ab}(z + 1/z)$, $h_z(z) = V + \delta_-(z - 1/z)$, and $h_0(z) = t_1(z + 1/z) + \delta_+(z - 1/z)$, with $\delta_\pm = (\delta_a \pm \delta_b)/2$. $H_{CNHSE-SSH}$ is so named because interestingly, at $\delta_- = \delta_{ab}$, it can be transformed via a basis rotation $\sigma_z \to \sigma_x$ into an extended SSH model with non-reciprocal inter-cell couplings given by $\pm 2\delta_-$ and a uniform non-reciprocal next-nearest neighbor hopping given by $t_1 \pm \delta_+$ (Supplementary Note 4), which is known to possess a $Z$-type topologically nontrivial phase.

When $\delta_{ab} = 0$, the system is decoupled into two Hatano–Nelson chains, which must be topologically trivial. The OBC spectrum $E_\infty$ in the decoupled case and the associated inverse decay length $\kappa$ are shown in Fig. 4b, c (green curves), with positive/negative $\kappa$ corresponding to skin modes accumulating population at opposite boundaries. Also shown in Fig. 4b, c (red curves) are $E_\infty$ in the coupled case and the corresponding $\kappa$ for the hybridized skin modes. With small $N = 20$ unit cells in

Fig. 4b, the finite-size OBC spectrum (gray dots) qualitatively agrees with the decoupled $E_\infty$ (green), with a real-valued gap at $E = 0$ along the $\mathrm{Im}[E] = 0$ axis (inset). Upon the size increase to $N = 30$ and then to $N = 40$, such a gap first closes on the complex plane and then develops into a point gap with two zero-energy degenerate modes lying in its center. The $Z$-type topological origin of such in-gap modes is also carefully verified in Supplementary Note 5. The gap closure and then the emergence of in-gap topological modes resemble the typical behavior of a topological phase transition. Yet, here it is an intriguing size-induced effect. Further, the emergence of in-gap modes only requires exponentially weaker inter-chain coupling (i.e., smaller $\delta_{ab}/\delta_a$) for larger $N$, as shown in the "phase" diagram shown in Fig. 4d.

**Proposal for circuit demonstration.** The CNHSE is most simply realized when the two subsystems have equal and opposite $\kappa$ values, since the system is then net reciprocal. Consider the RLC circuit as illustrated in Fig. 5. It is governed by Kirchhoff's law $\mathbf{I} = J\mathbf{V}$, where $\mathbf{I}$, $\mathbf{V}$ are the input currents and potentials at nodes 1A, 1B, 2A, 2B, ... and $J$ is the circuit Laplacian given, at AC frequency $\omega = (LC)^{-1/2}$, by

$$
\begin{aligned}
J(k) &= (2i\omega C \sin k)\sigma_z + r^{-1}(\mathbb{I} - \sigma_x) + 2R^{-1}(1 - \cos k)\mathbb{I} \\
&\rightarrow \begin{pmatrix} 2\omega C e^{ik} + \Delta(k) & -r^{-1} \\ -r^{-1} & 2\omega C e^{-ik} + \Delta(k) \end{pmatrix},
\end{aligned} \quad (5)
$$

where $\Delta(k) = r^{-1} + 2R^{-1} - 2(R^{-1} + \omega C)\cos k$. The second line was obtained via a unitary basis transformation $\sigma_y \rightarrow \tilde{\sigma}_y = U\sigma_y U^{-1} = \sigma_z$ that transforms the circuit Laplacian into a form similar to Eq. (3) (with $V = 0$), which is susceptible to the CNHSE. In this rotated basis, we evidently have two effective chains coupled by $-r^{-1}$, each with unbalanced gain/loss couplings that give rise to equal and opposite NHSE. Note that RLC components are all reciprocal and cannot realize the non-reciprocal effective chains individually. However, with the basis transformation given above, the two effective chains become entangled in a way such that they are net reciprocal and hence easy to realize with RLC components, as illustrated in Fig. 5.

One can experimentally demonstrate the CNHSE by building copies of the circuit with different numbers of unit cells $N$ (or alternatively by adjusting its length with appropriately placed switches), and mapping their Laplacian (admittance) spectra via established approaches[49–52]. For instance, one can systematically connect a current source $I$ to each node $\alpha$, one node at a time (the current exits through the ground), and measure the resultant electrical potentials $V_{\beta,\alpha}$ at each node $\beta$. The spectrum of $J$ is given by the inverse of the eigenvalues of the matrix $V_{\beta,\alpha}/I$. In the presence of the CNHSE, the spectral plots for different $N$ should qualitatively resemble that in Fig. 2b, since Eq. (5) is of the form of Eq. (3). Due to the robustness of the skin effect, component uncertainties in an actual experiment should minimally affect the resultant spectrum, as verified by simulation results presented in Supplementary Note 6. In particular, the circuit Laplacian spectra, which manifest the CNHSE, are almost undisturbed by uncertainty tolerances of up to 20%.

## Discussion

In mathematical terms, the CNHSE arises when the energy eigenequation exhibits an algebraic singularity that leads to inequivalent auxiliary GBZs across the transition. The CNHSE heralds a whole class of discontinuous critical phase transitions with rich anomalous scaling behavior, challenging traditional associations of criticality with scale-free behavior. Even a vanishingly small coupling between dissimilar skin modes can be consequential as the system size increases. This insight is much relevant to sensing and switching applications. Beyond our

two-chain models, there are other scenarios that can engineer coupling between subsystems of dissimilar NHSE length scales and hence yield CNHSE (e.g., see "Methods" section for a discussion of general two-band models). In particular, we anticipate fruitful investigations in various experimentally feasible settings such as electric circuits[53–56], cold atom systems[57,58], photonic quantum walks[59], and metamaterials[41,60], all of which are investigated with finite-size systems and hence highly relevant to the CNHSE.

## Methods

**Discontinuous transition of GBZ in two-chain models.** The discontinuous transition induced by an infinitesimal transverse coupling in the thermodynamic limit, and also the crossover in a finite system, exist only when the two decoupled chains have different $\kappa$ of their OBC skin solutions. To see this, we consider a general two-chain model described by Hamiltonian

$$
h(z) = \begin{pmatrix} g_a(z) + V_a & t_0 \\ t_0 & g_b(z) + V_b \end{pmatrix}, \quad (6)
$$

where $g_{a,b}(z)$ only contains terms with nonzero order of $z$. When decoupled, the two chains correspond to the polynomials $g_{a,b}(z) + V_{a,b}$, respectively, and possess the same $\kappa$ solutions when and only when $g_b(z) = cg_a(z)$, with $c$ a nonzero coefficient. When a nonzero transverse coupling $t_0$ is introduced, the characteristic polynomial of the two-chain system can always be written in the form of

$$
\begin{aligned}
P_c(z) &= (g_a(z) + V_a - E)(g_b(z) + V_b - E) - t_0^2 \\
&= (cg_a(z) - A)(g_a(z) - B),
\end{aligned} \quad (7)
$$

where $A$, $B$ are two coefficients determined by other parameters. Therefore for two chains with the same $\kappa$ solutions, a transverse coupling $t_0$ only modifies the energy offset between them, without inducing a transition of skin solutions.

Nevertheless, the above factorization does not hold when the coupling term $t_0$ is $z$-dependent, corresponding to inter-chain couplings between different unit cells. Under this condition, $P_c(z)$ cannot be factorized into two sub-polynomials of $g_a(z)$ and $g_b(z) = cg_a(z)$, meaning that the skin solution is changed for the system.

**GBZ solutions $E_\infty$ for the two-chain model.** For analytic tractability, we consider the case of Eq. 3 of the main text with $t_a^+ = t_b^- = 1$ and $t_a^- = t_b^+ = 0$ (i.e., $t_1 = \delta_a = -\delta_b = 0.5$), but nonzero $b$ and $V$. We obtain

$$
H_{2-\mathrm{chain}}(z) = \begin{pmatrix} z + V & t_0 \\ t_0 & 1/z - V \end{pmatrix}, \quad (8)
$$

with the characteristic polynomial given by

$$
\begin{aligned}
f(z, E) &= E^2 - E(z^{-1} + z) + [(z + V)(z^{-1} - V) - t_0^2] \\
&= \frac{V - E}{z} - z(V + E) + [E^2 - V^2 - t_0^2 + 1].
\end{aligned} \quad (9)
$$

To find the GBZ solutions $E_\infty$ for comparison with the actual OBC solutions, we solve for roots $|z_+| = |z_-|$ of $f(z, E) = 0$ (with $\Sigma = E^2 - V^2 - t_0^2 + 1$):

$$
\begin{aligned}
z_\pm &= \frac{\left(\Sigma \pm \sqrt{\Sigma^2 + 4(V^2 - E^2)}\right)}{2(V + E)} \\
&= \frac{\Sigma \pm \sqrt{(\Sigma - 2)^2 - 4t_0^2}}{2(V + E)}.
\end{aligned} \quad (10)
$$

For $|z_+| = |z_-|$ to hold, the square root quantity must differ from $\Sigma$ by a complex argument of $\pi/2$[38] i.e.,

$$
\sqrt{(\Sigma - 2)^2 - 4t_0^2} = i\eta\Sigma, \quad (11)
$$

where $\eta \in \mathbb{R}$. Simplifying, we obtain $\Sigma = \frac{2}{1 + \eta^2}\left(1 \pm \sqrt{t_0^2 + \eta^2(t_0^2 - 1)}\right)$ or, in terms of $E^2 \rightarrow E_\infty^2$,

$$
E_\infty^2 = \frac{1 - \eta^2 \pm 2\sqrt{t_0^2 - \eta^2 + \eta^2 t_0^2}}{1 + \eta^2} + V^2 + t_0^2. \quad (12)
$$

as in the main text, with $\eta$ tracing out a one-parameter continuous spectrum. The GBZ can be numerically obtained by substituting Eq. (12) into the expression for $z_\pm$ in Eq. (10) with $E = E_\infty$. From that, we obtain two momentum values $k_\pm = \mathrm{Re}[-i\log z_\pm]$ with $\kappa(k_+) = \kappa(k_-) = -\log|z_+| = -\log|z_-|$ as the inverse length scales. Note however that because of the proximity to the $t_0 = 0$ critical point, this value of $\kappa(k_\pm)$ is significantly different from the actual inverse OBC skin depth for a large range of finite system sizes.

**Dissimilar skin modes in general two-band models.** In a more general picture, the CNHSE and the size-dependent variation may exist when different parts of the system have dissimilar skin accumulation of eigenmodes. In the two-chain model, we mainly consider regimes with small inter-chain couplings, thus the two energy bands (overlapped or connected in most cases) with dissimilar skin modes are mostly given by one of the two chains respectively. To unveil the condition of having dissimilar skin modes in a general two-band system, we consider an arbitrary two-band

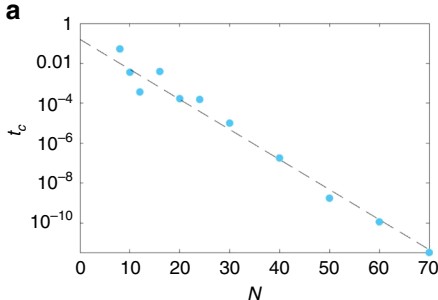
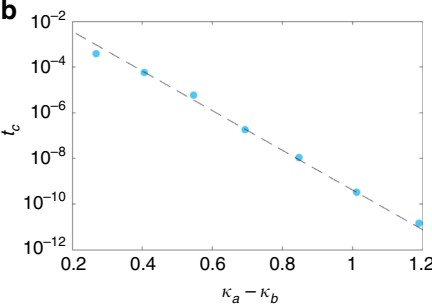

**Fig. 6 Inverse exponential scaling of the critical bare coupling $t_0 = t_c$ required for the open-boundary spectrum of $H_{2\text{-chain}}$ to make the transition from real to complex. a, b** $t_c$ Versus the system's size $N$ and effective skin depth $\kappa_a - \kappa_b$, respectively. The numerical data (blue) fits very well with the predicted scaling law $t_c \sim e^{-(\kappa_a - \kappa_b)N/2}$ (dashed lines) with $\kappa_a - \kappa_b = \log 2$ in **a** and $N = 40$ in **b**. Unless specified in the figure, the parameters are $t_1 = 0.75$, $\delta_a = -\delta_b = 0.25$ as in Fig. 2 of the main text. In **b**, $\kappa_a - \kappa_b$ is obtained from Eq. (14) with $\delta_a = -\delta_b$ varying from 0.1 to 0.4.

system described by a non-Bloch Hamiltonian $H(z) = h_0(z)\mathbb{I} + \sum_{n=1,2,3} h_n(z)\sigma_n$, with $z = e^{ik}e^{-\kappa(k)}$, and $\kappa(k)$ a complex deformation of momentum $k$ describing the NHSE. Its characteristic polynomial is given by

$$f(z, E) = [E - h_0(z)]^2 - P(z) = 0, \tag{13}$$

with $P(z) = \sum_{n=1,2,3} h_n^2(z)$. NHSE can be described by a GBZ where the solutions of $f(z, E) = 0$ satisfy $E_\alpha(z_\mu) = E_\alpha(z_\nu)$ with $|z_\mu| = |z_\nu|$ and $\alpha = \pm$ the band index, and $\kappa(k) = -\log|z|$ gives the inverse decay length. Conventionally, NHSE is studied mostly for a system with only nonzero $h_0(z)$ (i.e., a one-band model) or $P(z)$ (e.g., the non-reciprocal SSH model), where the zeros of $f(z, E)$ lead to $E_\pm = h_0(z)$ and $E_\pm^2 = P(z)$, respectively. In either case, we can see that the two bands of $E_\pm$ must have the same inverse skin localization depth $\kappa(k)$, as $E_\alpha(z_\mu) = E_\alpha(z_\nu)$ must be satisfied for $\alpha = \pm$ with the same $z_{\mu,\nu}$. To dissimilar skin modes for the two bands, $h_0(z)$ and $P(z)$ must both be non-vanishing, and possess different skin solutions. That is, although $h_0(z_\mu) = h_0(z_\nu)$ and $P(z_{\mu'}) = P(z_{\nu'})$ can still be satisfied with $|z_\mu| = |z_\nu|$ and $|z_{\mu'}| = |z_{\nu'}|$, we cannot have $z_\mu = z'$ and $z_\nu = z'$ at the same time, otherwise the same $\kappa(k)$ can be obtained for the two bands.

**Competition between skin localization and inter-chain coupling.** As mentioned in the main text, if two coupling chains have inverse NHSE decay lengths (non-Hermitian localization length scales) $\kappa_a$, $\kappa_b$, a change of basis will bring their coupling to be effective between a chain with no NHSE, and another with an effective skin depth $\kappa_a - \kappa_b$. Since that entails exponentially growing skin modes scaling like $e^{(\kappa_a - \kappa_b)N}$ at one end, we expect the effect of even an infinitesimally small inter-chain coupling $t_0$ to scale exponentially with $N$, and eventually change the OBC spectrum substantially.

Consider increasing the inter-chain coupling $t_0$ in our two-chain model (Eq. 3 of main text) from zero. At sufficiently small $t_0$, we have two practically independent OBC Hatano–Nelson chains with real spectra. Their infinitesimal coupling only shifts their eigenenergies slightly along the real line. But at a critical $t_0 = t_c$, the OBC spectrum is rendered complex as one or more pairs of eigenenergies coalesce and repel along in the imaginary direction. Shown in Fig. 6a is the inverse exponential scaling of the critical $t_0 = t_c$ with $N$. We observe that $t_c^2 e^{(\kappa_a - \kappa_b)N} \sim \mathcal{O}(1)$, in agreement with the intuitive expectation that $t_c$ should scale inverse exponentially with $N$ because the effect of $t_0$ scales exponentially with $N$. Yet, the fact that $t_c^2 \sim e^{-(\kappa_a - \kappa_b)N}$ signifies that the CNHSE is fundamentally a non-perturbative effect since it differs from $t_c \sim e^{-(\kappa_a - \kappa_b)N}$ as expected from first-order perturbation theory with left and right eigenstates that are oppositely exponentially localized spatially.

The scaling behavior of $e^{(\kappa_a - \kappa_b)N}$ also suggests that increasing $N$ has similar consequences as increasing the non-reciprocity in the system, the strength of which is reflected by the absolute value of $(\kappa_a - \kappa_b)$. Therefore it is also expected that the CNHSE shall emerge when we enhance the non-reciprocity but fix $N$. In Fig. 6b, we show the inverse exponential scaling of the critical $t_0 = t_c$ with $\kappa_a - \kappa_b$, where the inverse NHSE decay lengths are given by

$$e^{\kappa_{a,b}} = \sqrt{\frac{t_1 + \delta_{a,b}}{t_1 - \delta_{a,b}}} \tag{14}$$

for the two decoupled chains. The scaling behavior versus $\kappa_a - \kappa_b$ further confirms that $t_c^2 \sim e^{-(\kappa_a - \kappa_b)N}$.

## Data availability
Raw numerical data from the plots presented are available from the authors upon request.

## Code availability
Though not essential to the central conclusions of this work, computer codes for generating our figures are available from L.L. and C.H.L. upon reasonable request.

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

## Acknowledgements

We thank Nobuyuki Okuma and Zhesen Yang for helpful discussions. J.G. acknowledges support from Singapore NRF Grant No. NRF-NRFI2017-04 (WBS No. R-144-000-378-281).

## Author contributions

L.L. and C.H.L. contributed equally to this work. L.L. carried out preliminary studies and all authors participated in the discussions. S.M. helped to improve the design of lattice models. C.H.L. refined this project extensively. L.L. and C.H.L. carried out additional theoretical and computational studies. All authors discussed the results and participated in the writing of the manuscript. J.G. supervised the project and finalized the manuscript.

## Competing interests

The authors declare no competing interests.
