## [Peer Review File · Nature Communications]

REVIEWER COMMENTS

Reviewer #1 (Remarks to the Author):

In the submitted manuscript titled "Critical non-Hermitian skin effect", Li et al investigated the critical behaviors of non-Hermitian spectra near the critical points. The study of non-Hermitian systems is a very vigorous field that has attracted growing attention recently. Li et al found a new class of critical behaviors that is unique to non-Hermitian systems. Specifically, they found that two non-Hermitian systems with different generalized Brillouin zone (GBZ) is generally critical at zero coupling, and adding a small coupling causes intriguing critical behaviors. This is a very interesting piece of work that is expected to have broad interests to the field of non-Hermitian physics. Therefore, I would like to enthusiastically recommend its publication in Nature Communications, provided that the following issues could be satisfactorily addressed:

- 1) The presentation is somewhat confusing regarding the role of GBZ. The authors present the results as a "failure" of the GBZ picture. However, what I can see here is an illuminating application of GBZ, which shows that the GBZ picture is not only valid, but even more powerful than naively expected. Indeed, the GBZ produces the continuous bulk spectra as long as the size is sufficiently large; and this is perfectly valid here. I think it is more illuminating to present the critical non-Hermitian skin effect as a GBZ-enforced critical phenomena: Because the GBZ at zero coupling and an infinitesimal coupling are drastically different, the continuous energy spectra (at large system-size) must change discontinuously in the presence of a tiny coupling. Meanwhile, since small-size system cannot feel such drastic change when the coupling is tiny, the GBZ discontinuity guarantees that interesting behaviors must exist when the size grows from small to large. I think that this logic of GBZ-enforced critical phenomena is much clearer than describing the results as a failure of GBZ, which works perfectly here. The result is a nontrivial and insightful application of GBZ. As such, I suggest a revision of the relevant descriptions.
- 2) It would be helpful to include in the main text a figure of the GBZs at zero coupling and at an infinitesimal coupling.
- 3) The GBZ equation $z = (+-)\sqrt{V^2 + \exp(i\theta)} - V$ looks incomplete. In fact, I reproduced this calculation and found that this is only part of the GBZ; the missing part is $z = 1/\{V(+)-\sqrt{V^2 + \exp(i\theta)}\}$. These two parts sum to the entire GBZ.
- 4) "Critical skin effect (CSE)" should be modified to "critical non-Hermitian skin effect (cNHSE)" throughout the paper. The word "non-Hermitian" should be retained to distinguish it from the skin effect in electromagnetism.
- 5) The third row of Fig.1 is somewhat misleading. I can understand what the authors mean, however, I am afraid that many readers would see it as an abrupt change in spectra (from several line segments to two points), just like the abrupt change illustrated in the fourth row. Therefore, I suggest either to improve the third row or simply to remove it.

Reviewer #2 (Remarks to the Author):

I have read the manuscript "Critical non-Hermitian Skin Effect" with interest. In non-Hermitian systems with unbalanced gain and loss, the spectra under periodic boundary conditions and open boundary conditions can be different. This leads to such major effects as the non-Hermitian skin effect characterized by enigmatic bulk-boundary correspondence violations. In the previous works, a generalized Brillouin zone approach is widely used to describe the non-Hermitian systems and define topological invariants leading to a general bulk-boundary correspondence, as one can compute the open boundary spectrum of systems of large or infinite size. The large or infinite size is a condition for the application of a generalized Brillouin zone approach. Thus it is not surprising that the generalized Brillouin zone approach is not suitable for finite systems.

While I find the logic of the work is clear, and have no reason to doubt the numerical results, I do disagree with the authors on the relevance of the results.

I agree that non-Hermitian skin effect and non-Hermitian bulk-boundary correspondence are crucial and

important for us to understand non-Hermitian systems, but given the nature of the non-Hermitian system and its dynamics, I do not quite see that this importance will carry over for a system of finite size. The results in the current manuscript are much less fundamental compared to the previous ones in which analytical or asymptotical results are given for systems of large or infinite size. Indeed, I would argue that the lack of investigation of numerical results of physical phenomena of non-Hermitian systems with finite size is not due to a difficulty preventing other researchers from studying it, but simply due to a lack of the relevance, which make them less interesting to study. I would recommend to publish the paper in a more specialized journal then.

There are two minor additional comments. 1 There are a few typos and punctuations such as period missing. 2 It is a little vague how to demonstrate the so-called critical non-Hermitian skin effect in the RLC circuit. I would strongly recommend to emphasize this more, since otherwise the experimental results might appear very different from that are expected in this manuscript (as shown in Ref. [53]).

Reviewer #3 (Remarks to the Author):

In this paper, the authors investigate a phenomenon, where a small perturbation in the system leads to a discontinuous transformation of the energy spectrum and eigenstates, which they call the critical non-Hermitian skin effect. They show that an approach based on defining a generalized Brillouin zone, which was introduced by Yao and Wang in 2018 (Ref.26) to remedy the possible breakdown of the bulk-boundary correspondence in non-Hermitian models, breaks down for models featuring such behavior, and that this breakdown occurs when the characteristic equation for the Hamiltonian becomes irreducible. To illustrate these properties, the authors study examples of two coupled non-Hermitian one-dimensional Hatano-Nelson chains, and find that the properties of the open-boundary-condition spectrum are dependent on the system size. They also find that the skin modes, i.e., the bulk modes that pile up at the boundaries, display scale-free power-law behavior, and they observe crossover behavior in the entanglement entropy. Additionally, the authors propose an experimental realization of their model in electric circuits.

Non-Hermitian topological phases are currently a very popular research subject, and attract much attention due to their exotic properties such as the breakdown of bulk-boundary correspondence accompanied by the piling up of bulk states at the boundaries, which is central to the content of this manuscript. The findings in this work, namely, the occurrence of a discontinuous transition in the behavior of the spectrum and eigenstates upon turning on a small coupling term, are quite interesting and novel, and an important contribution to the field of non-Hermitian models. The results seem to be valid, and should be straightforwardly reproducible by others. The fact that the method of defining a generalized Brillouin zone breaks down for models with such behavior is an interesting and important result, as many works rely on this approach.

While the paper is generally well written, there are some small grammatical mistakes here and there. Additionally, I have the following questions and comments for the authors:

1. Ref.23 came after Okugawa and Yokoyama, PRB 99, 041202(R) (2019) and Budich et al., PRB 99, 041406(R) (2019), which also focus on symmetry considerations and exceptional structures. Also, for completeness, the authors should cite Kunst et al., PRL 121, 026808 (2018) because this paper together with Ref.26 (Yao and Wang) was among the first to study and remedy the breakdown of bulk-boundary correspondence. Lastly, the link between the circuit Laplacian and the impedance is also already established in Ningyan et al., PRX 5, 021031 (2015), and should be acknowledged.
2. The authors study the entanglement entropy in Fig.3(b) and find crossover behavior around $N = 5$. For completeness, the authors should explain the meaning of going from zero to what they call the usual behavior, i.e., scaling with $1/3 \log N$.
3. In the caption of Fig.3, the authors write that they consider odd system sizes N with a cut at $N/2$. But if N is odd, $N/2$ is a fractional number. Is this indeed what they mean?
4. I am a bit confused about what the authors mean with "the real OBC spectrum E_{∞} ". Do they mean the actual spectrum when diagonalizing the Hamiltonian matrix, or do they mean the real part of the spectrum?
5. In the caption of figure 4, the authors say that the green and blue lines correspond to the skin solutions.

In panel (c), I can only see green lines no blue lines.

6. In the main body of the paper and in the supplementary material, the authors make use of a rotation of basis in the Bloch Hamiltonian to change their models. However, seeing the exotic properties displayed by these models, how can they be sure that these models, which are linked via such a rotation, behave the same under open boundary conditions?

Detailed below are our responses to the comments of the reviewers, together with a description of the associated changes in the manuscript.

Response to Reviewer #1

In the submitted manuscript titled “Critical non-Hermitian skin effect”, Li et al investigated the critical behaviors of non-Hermitian spectra near the critical points. The study of non-Hermitian systems is a very vigorous field that has attracted growing attention recently. Li et al found a new class of critical behaviors that is unique to non-Hermitian systems. Specifically, they found that two non-Hermitian systems with different generalized Brillouin zone (GBZ) is generally critical at zero coupling, and adding a small coupling causes intriguing critical behaviors. This is a very interesting piece of work that is expected to have broad interests to the field of non-Hermitian physics. Therefore, I would like to enthusiastically recommend its publication in Nature Communications, provided that the following issues could be satisfactorily addressed:

We are pleased that Reviewer #1 finds our work very interesting with broad interest within the very vigorous field of non-Hermitian physics, and enthusiastically recommends it for publication in Nature Communications. We are also grateful to him/her for the following suggestions for improvement, which we have fully adopted in our revision.

1) The presentation is somewhat confusing regarding the role of GBZ. The authors present the results as a “failure” of the GBZ picture. However, what I can see here is an illuminating application of GBZ, which shows that the GBZ picture is not only valid, but even more powerful than naively expected. Indeed, the GBZ produces the continuous bulk spectra as long as the size is sufficiently large; and this is perfectly valid here. I think it is more illuminating to present the critical non-Hermitian skin effect as a GBZ-enforced critical phenomena: Because the GBZ at zero coupling and an infinitesimal coupling are drastically different, the continuous energy spectra (at large system-size) must change discontinuously in the presence of a tiny coupling. Meanwhile, since small-size system cannot feel such drastic change when the coupling is tiny, the GBZ discontinuity guarantees that interesting behaviors must exist when the size grows from small to large. I think that this logic of GBZ-enforced critical phenomena is much clearer than describing the results as a failure of GBZ, which works perfectly here. The result is a nontrivial and insightful application of GBZ. As such, I suggest a revision of the relevant descriptions.

We are grateful to our reviewer for providing this insightful comment regarding the role of GBZ in our study. As our reviewer would agree, for an extremely large system, the GBZ solutions faithfully reflect the spectra with and without a vanishingly small inter-chain coupling, and predict the discontinuous change of the spectrum due to the difference between the two GBZ solutions. In a finite-size system of practical interest, it is also the divergence between the two GBZ solutions that enforces the smooth crossover between them. That is, a finite-size system with a tiny inter-chain coupling falls in the regime of the crossover, meaning that its spectrum and eigenstates are different from either GBZ solution. Nevertheless, we perfectly agree with our reviewer that these observations do not reflect a “failure” of the GBZ picture, which was meant for systems in the thermodynamic limit after all. Indeed, it is the insights of the GBZ picture that help us understand the interesting behavior in the crossover regime for finite-size systems.

In our previous text, we used phrases such as failure of GBZ for finite-size systems. Thanks to our reviewer’s comment, we find that such phrases, though still correct, will probably cause misunderstanding. Following our reviewer’s suggestion, we have revised some relevant descriptions (including changes in the abstract) in our revised version, which shall make our discussion more accurate regarding the GBZ picture.

2) It would be helpful to include in the main text a figure of the GBZs at zero coupling and at an infinitesimal coupling.

We thank our reviewer for this very nice suggestion. As we have revised some relevant discussions about the GBZ, it indeed becomes more important to show the GBZs at different coupling conditions, as suggested by our reviewer. Hence we have now added a panel (c) in Fig. 2 illustrating the GBZs of the system at two different couplings, with necessary discussions in the revised main text.

3) The GBZ equation $z = (\pm) \sqrt{V^2 + \exp(i\theta)} - V$ looks incomplete. In fact, I reproduced this calculation and found that this is only part of the GBZ; the missing part is $z = 1 / \{V(\pm) \sqrt{V^2 + \exp(i\theta)}\}$. These two parts sum to the entire GBZ.

We are sorry for overlooking this. We are grateful to our reviewer for correctly pointing out this missing part. We have now corrected it in the revised version.

4) “Critical skin effect (CSE)” should be modified to “critical non-Hermitian skin effect (cNHSE)” throughout the paper. The word “non-Hermitian” should be retained to distinguish it from the skin effect in electromagnetism.

We have changed this terminology accordingly throughout the paper.

5) The third row of Fig.1 is somewhat misleading. I can understand what the authors mean, however, I am afraid that many readers would see it as an abrupt change in spectra (from several line segments to two points), just like the abrupt change illustrated in the fourth row. Therefore, I suggest either to improve the third row or simply to remove it.

Thanks again for another valuable suggestion. To make it clearer that the non-Hermitian gapped transition and the other two gapped transitions are continuous transitions, we have added several arrows to indicate how the spectra vary across the transition point. We have also further emphasized in the figure caption that row 3 refers to a continuous deformation but not row 4. We hope that these changes have considerably improved this figure.

Response to Reviewer #2

I have read the manuscript "Critical non-Hermitian Skin Effect" with interest. In non-Hermitian systems with unbalanced gain and loss, the spectra under periodic boundary conditions and open boundary conditions can be different. This leads to such major effects as the non-Hermitian skin effect characterized by enigmatic bulk-boundary correspondence violations. In the previous works, a generalized Brillouin zone approach is widely used to describe the non-Hermitian systems and define topological invariants leading to a general bulk-boundary correspondence, as one can compute the open boundary spectrum of systems of large or infinite size. The large or infinite size is a condition for the application of a generalized Brillouin zone approach. Thus it is not surprising that the generalized Brillouin zone approach is not suitable for finite systems.

While I find the logic of the work is clear, and have no reason to doubt the numerical results, I do disagree with the authors on the relevance of the results.

I agree that non-Hermitian skin effect and non-Hermitian bulk-boundary correspondence are crucial and important for us to understand non-Hermitian systems, but given the nature of the non-Hermitian system and its dynamics, I do not quite see that this importance will carry over for a system of finite size. The results in the current manuscript are much less fundamental compared to the previous ones in which analytical or asymptotical results are given for systems of large or infinite size. Indeed, I would argue that the lack of investigation of numerical results of physical phenomena of non-Hermitian systems with finite size is not due to a difficulty preventing other researchers from studying it, but simply due to a lack of the relevance, which make them less interesting to study. I would recommend to publish the paper in a more specialized journal then.

Clearly, our second reviewer's main reservation is that finite-size non-Hermitian systems are less interesting due to a lack of relevance. While we fully respect the viewpoint of the reviewer, below we would like to explain why we think our work is extremely relevant to all experimental and numerical studies of a broad class of critical non-Hermitian systems, as well as how it is of fundamental significance.

First of all, our results are applicable not just to small systems, but actually applies to arbitrarily large finite-sized systems. Arguably, larger systems are even more susceptible to the critical skin effect, because any non-vanishing coupling can induce the anomalous effects of such criticality at sufficiently large size. As such, our results are relevant to all experiments and numerical simulations of such classes of non-Hermitian systems, since physical experiments and simulations must always be performed on a finite lattice, even if their goal is to illustrate properties that hold in the infinite size limit. That includes various existing experimental setups that have realized generalized Brillouin zone skin effect properties, e.g. quantum walks of single photons [1], classical circuit lattices [2,3], and metamaterials [4,5].

Because all available experiments and numerical simulations are carried out on finite-size systems, we wish to highlight here that, there is no "lack of investigation" or "lack of relevance" of finite-size non-Hermitian systems at all. In our view, what is truly lacking is our new knowledge about exotic/critical behaviors in finite-size non-Hermitian systems (however large they are). That is, precisely building on previous analytical or asymptotical results on GBZ, which are in-principle for systems of infinite size, we are now forced to accept that a finite system involving a vanishingly small coupling between two GBZs, however large the system's size is, may not be faithfully described by a GBZ picture.

As we elaborate in the manuscript, the criticality induced by a small inter-chain coupling is reflected by a discontinuous “jump” of the spectrum and eigenstates across the critical point in the thermodynamic limit, which smoothens out into a crossover between different phases for finite-size systems. In other words, for an arbitrarily small perturbation, the finite-size effect can occur in a sufficiently large system. This extreme sensitivity to perturbations we unveil provides a useful mechanism for sensing applications, different from that of another recent preprint appeared after our paper [6].

Besides being universally relevant in experiments and simulations, the anomalous size dependence in the critical skin effect is of fundamental significance because it challenges a number of phenomena that are often taken for granted in other critical systems. For instance, in our context, scale-free behavior can be consistent with exponential decay, the scaling of entanglement entropy can depend on a length scale even in the presence of criticality, the un-exchangeable two limits of thermodynamic limit and zero-coupling limit, the transition that looks like a phase transition, but actually is due to a size change etc. Implied in it is a challenge to the notion of the thermodynamic limit itself – in extrapolating a generic system exhibiting the critical skin effect to very large sizes, the system may necessarily pass through qualitatively different regimes, appearing to converge to their respective “thermodynamic limit” phases for a while, before transiting into other qualitatively different phases. Indeed, the success of the theory of the GBZ makes many people naively believe that so long as the system size is quite large, it would be ok. But with results from the critical skin effect, one realizes that the requisite system size may be unusually large depending on the coupling, and as shown in this work (which may go unnoticed by our reviewer), this dependence is logarithmically diverging with the coupling strength.

Finally, we do agree with our 2nd reviewer that there is no technical difficulty in investigating finite-size non-Hermitian systems. That there are not many serious discussions on exotic finite-size effect are not because of the lack of relevance, but that the critical behavior and its finite-size variation have not been noticed yet, or that a proper description of them has not yet been found, as the NHSE and the GBZ picture that our study is built on are unveiled less than 2 years ago.

To end our exchanges with 2nd reviewer that have given us more confidence about the importance of our work, we would like to point out that the significance of our work is also clearly acknowledged by the other two reviewers. We thus wish to ask our 2nd reviewer to reconsider our manuscript for publication in Nature Communications. We have also revised our manuscript to make the importance of our work clearer to the readers.

- [1] L. Xiao, T. Deng, K. Wang, et al., Nature Physics , 1–6 (2020).
- [2] T. Helbig, T. Hofmann, S. Imhof, et al., Nature Physics , 1–4 (2020).
- [3] T. Hofmann, T. Helbig, F. Schindler, et al., Physical Review Research 2, 023265 (2020).
- [4] M. Brandenbourger, X. Locsin, E. Lerner, and C. Coulais, Nature communications 10, 1–8 (2019).
- [5] A. Ghatak, M. Brandenbourger, J. van Wezel, and C. Coulais, arXiv: 1907.11619. .
- [6] J. C. Budich and E. J. Bergholtz arXiv: 2003.13699.

There are two minor additional comments.

1 There are a few typos and punctuations such as period missing.

We are grateful to our reviewer for his/her careful reading of the manuscript. During the revision we have proof-read the whole manuscript a few more times and have corrected several typos and punctuations in the revised version.

2 It is a little vague how to demonstrate the so-called critical non-Hermitian skin effect in the RLC circuit. I would strongly recommend to emphasize this more, since otherwise the experimental results might appear very different from that are expected in this manuscript (as shown in Ref. [53]).

We fully agree with our reviewer that it is important to show that a real experiment should yield results that agree well with the theoretical predictions in this manuscript. Following our reviewer's advice, in the revised manuscript we have added a new paragraph describing how such circuits can be built and their admittance spectra of their Laplacian measured.

Although circuits can replicate the structure of tight-binding models exactly, inevitable component uncertainties arise in any actual implementation, and that can potentially lead to different measured Laplacian spectra. Hence we have additionally performed simulations where all components are assumed to have component uncertainty tolerances of 5%, which is reasonable in practice, and 20%, which may be encountered in low-cost experiments. As presented in a new subsection in the Supplementary Information, we indeed do not observe any significant change in the spectrum even under 20% uncertainty tolerances. This is due to the robustness of the skin effect in general, which causes the skin modes to accumulate in the same way as long as there are no drastic impurity hoppings that reverse the direction of accumulation. As such, we concluded that our CNHSE can indeed be convincingly demonstrated in a proposed circuit experiment using circuit components of very reasonable and easily procurable quality.

Response to Reviewer #3

In this paper, the authors investigate a phenomenon, where a small perturbation in the system leads to a discontinuous transformation of the energy spectrum and eigenstates, which they call the critical non-Hermitian skin effect. They show that an approach based on defining a generalized Brillouin zone, which was introduced by Yao and Wang in 2018 (Ref.26) to remedy the possible breakdown of the bulk-boundary correspondence in non-Hermitian models, breaks down for models featuring such behavior, and that this breakdown occurs when the characteristic equation for the Hamiltonian becomes irreducible. To illustrate these properties, the authors study examples of two coupled non-Hermitian one-dimensional Hatano-Nelson chains, and find that the properties of the open-boundary-condition spectrum are dependent on the system size. They also find that the skin modes, i.e., the bulk modes that pile up at the boundaries, display scale-free power-law behavior, and they observe crossover behavior in the entanglement entropy. Additionally, the authors propose an experimental realization of their model in electric circuits.

Non-Hermitian topological phases are currently a very popular research subject, and attract much attention due to their exotic properties such as the breakdown of bulk-boundary correspondence accompanied by the piling up of bulk states at the boundaries, which is central to the content of this manuscript. The findings in this work, namely, the occurrence of a discontinuous transition in the behavior of the spectrum and eigenstates upon turning on a small coupling term, are quite interesting and novel, and an important contribution to the field of non-Hermitian models. The results seem to be valid, and should be straightforwardly reproducible by others. The fact that the method of defining a generalized Brillouin zone breaks down for models with such behavior is an interesting and important result, as many works rely on this approach.

While the paper is generally well written, there are some small grammatical mistakes here and there. Additionally, I have the following questions and comments for the authors:

We are very glad and feel greatly encouraged that our 3rd reviewer found our findings “interesting and novel, and an important contribution to the field of non-Hermitian models”, and furthermore “should be straightforwardly reproducible by others”. In our revised manuscript, we have also added a new paragraph and a supplementary subsection that elaborates on the circuit proposal for experimental reproducibility.

We also thank our reviewer for carefully reading our manuscript and for pointing out some minor language mistakes. We have now proof-read our manuscript with great care and have corrected these.

1. Ref.23 came after Okugawa and Yokoyama, PRB 99, 041202(R) (2019) and Budich et al., PRB 99, 041406(R) (2019), which also focus on symmetry considerations and exceptional structures. Also, for completeness, the authors should cite Kunst et al., PRL 121, 026808 (2018) because this paper together with Ref.26 (Yao and Wang) was among the first to study and remedy the breakdown of bulk-boundary correspondence. Lastly, the link between the circuit Laplacian and the impedance is also already established in Ningyan et al., PRX 5, 021031 (2015), and should be acknowledged.

We are grateful to our highly knowledgeable reviewer for reminding us of these related references. We have now added them in the revised version. Nature Communications has a limit in the number of references, and this sometimes forced us the authors to be unnecessarily biased in selecting the references.

2. The authors study the entanglement entropy in Fig.3(b) and find crossover behavior around $N = 5$. For completeness, the authors should explain the meaning of going from zero to what they call the usual behavior, i.e., scaling with $1/3 \log N$.

We thank again our reviewer for this valuable suggestion. Entanglement entropy (EE) of gapped fermionic systems obeys an area or boundary law, which states that EE is proportional to the size of the boundary of the system. In one dimension (1D), the size of the boundary is independent on the system length, therefore we observe a saturation of entanglement entropy below $N=5$ when the system is gapped. This area law is violated for 1D conformal theories, e.g. gapless free fermions, and the scaling of EE is replaced by a logarithm dependence $\sim \log(N)$ on the system size, and the coefficient in front of the logarithm is universal and proportional to the central charge of the conformal field theory. In summary, the transition from a gapped system to a gapless system gives rise to an unusual EE scaling for free fermions, which cannot be described by a single well-defined scaling law.

Following our reviewer's advice, we have now revised some corresponding descriptions in the manuscript to explain this unusual scaling behavior of EE more clearly.

3. In the caption of Fig.3, the authors write that they consider odd system sizes N with a cut at $N/2$. But if N is odd, $N/2$ is a fractional number. Is this indeed what they mean?

This question can be easily addressed. We actually considered a cut at $[N/2]$ (as already stated in our previous version through the symbol $[]$), i.e. a floor function of $N/2$. This is now further clarified in the caption of Fig. 3 of the revised version.

4. I am a bit confused about what the authors mean with "the real OBC spectrum E_{∞} ". Do they mean the actual spectrum when diagonalizing the Hamiltonian matrix, or do they mean the real part of the spectrum?

We are sorry for being unclear on this. We mean the actual spectrum, which takes real values under OBC. In the caption for Fig. 2, we have rephrased this statement to eliminate this unfortunate ambiguity. There we now use "the real-value OBC spectrum" to clarify.

5. In the caption of figure 4, the authors say that the green and blue lines correspond to the skin solutions. In panel (c), I can only see green lines no blue lines.

We are very sorry for this mistake generated during the editing of our manuscript. There is indeed no blue line in the figure (all relevant lines are green). We have now corrected the descriptions in the revised version.

6. In the main body of the paper and in the supplementary material, the authors make use of a rotation of basis in the Bloch Hamiltonian to change their models. However, seeing the exotic properties displayed by these models, how can they be sure that these models, which are linked via such a rotation, behave the same under open boundary conditions?

This is a very elegant question. We note that the rotation is only applied to the basis of pseudospin (Pauli matrices). For an OBC system, this rotation can be simply represented by a unitary transformation of the real-space Hamiltonian, which only redefines the two sublattices within each unit cell, but does not change the eigen-energies and the spatial profile of the eigenstates. We have also verified this by numerically diagonalizing the Hamiltonians before and after the rotation, which produce almost identical results, with numerical error of eigen-energies at the scale of $\sim 10^{-15}$ for $N=40$. In the revised version, we have added a sentence to clarify this point when first introducing

such a rotation in the Supplementary Materials. Also note that this rotation of basis is different from the reorganization of unit cells discussed around Eq. S9 in the supplementary information.

With these exchanges and necessary changes described above, we hope that we have now satisfactorily addressed all the concerns from our 3rd reviewer.

.

REVIEWERS' COMMENTS:

Reviewer #1 (Remarks to the Author):

I think that all the issues raised in the previous reports have been satisfactorily addressed, and I strongly recommend the revised manuscript to Nature Communications.

About the comments from the 2nd reviewer. I do agree with the 2nd referee that infinite-size limit is the most fundamental to non-Hermitian topology. Indeed, even for the models considered here, the energy spectrums still converge to the predictions of generalized Brillouin zone when the system size is sufficiently large. Nevertheless, I think that this paper points out a highly interesting non-Hermitian phenomenon, namely that the generalized Brillouin zone can jump discontinuously at certain point in the parameter space, and that these jumps in GBZ have very interesting physical consequences such as novel scaling behaviors in finite size systems. This is an elegant application of the generalized Brillouin zone, and the "critical non-Hermitian skin effect" predicted by the GBZ discontinuity is a unique non-Hermitian phenomenon. I agree with the authors that this piece of physics had not noticed before not because of its lack of relevance, but because the generalized Brillouin zone and non-Bloch topology have been unveiled only two years ago. It looks that there remains much physics hidden in the non-Hermitian topology (and particularly, in the generalized Brillouin zone). I think the authors was partially responsible for the misunderstanding because the initially submitted manuscript describes the results as a "failure" of GBZ, which was misleading. In fact, the physics is an illuminating application of GBZ. The revised manuscript is now much clearer.

Reviewer #2 (Remarks to the Author):

The revised manuscript has addressed two minor concerns in my first report. However, I insist that the results shown in this manuscript are much less fundamental compared to the previous ones in which analytical or asymptotical results are given for systems of large or infinite size. The reply to this major concern is not convincing. The authors claimed that their results are relevant to all experiments and numerical simulations of such classes of non-Hermitian systems, since physical experiments and simulations must always be performed on a finite lattice. Theories should be guides and predict the physical phenomena. I do not understand the point raised by the authors that finite-size non-Hermitian systems are fundamental as all available experiments and numerical simulations are carried out on finite-size systems. Furthermore, the experiments in references [1-4] are all proof-of-principle experiments and try to prove the previous theories applied on physical systems with large or infinite sizes. Therefore, I still do not see the necessity and importance of the study carrying over for a system of finite size.

Reviewer #3 (Remarks to the Author):

I am satisfied with the responses of the authors to the referee reports, and the improvements made to the paper. I believe this work is now ready for publication.

One small comment: At the bottom of page 4, the authors refer to Fig.3(c). This should be Fig.3(b).